# TMEM16B regulates anxiety-related behavior and GABAergic neuronal signaling in the central lateral amygdala

Ke-Xin Li[1,2,3], Mu He[1,2,3], Wenlei Ye[1,2,3], Jeffrey Simms[4], Michael Gill[4], Xuaner Xiang[5], Yuh Nung Jan[1,2,3], Lily Yeh Jan[1,2,3]*

[1]Department of Physiology, University of California, San Francisco, San Francisco, United States; [2]Department of Biochemistry and Biophysics, University of California, San Francisco, San Francisco, United States; [3]Howard Hughes Medical Institute, University of California, San Francisco, San Francisco, United States; [4]Gladstone Institute of Neurological Disease, San Francisco, United States; [5]Department of Anesthesia and Perioperative Care, University of California, San Francisco, San Francisco, United States

**Abstract** TMEM16B (ANO2) is the $Ca^{2+}$-activated chloride channel expressed in multiple brain regions, including the amygdala. Here we report that *Ano2* knockout mice exhibit impaired anxiety-related behaviors and context-independent fear memory, thus implicating TMEM16B in anxiety modulation. We found that TMEM16B is expressed in somatostatin-positive (SOM[+]) GABAergic neurons of the central lateral amygdala (CeL), and its activity modulates action potential duration and inhibitory postsynaptic current (IPSC). We further provide evidence for TMEM16B actions not only in the soma but also in the presynaptic nerve terminals of GABAergic neurons. Our study reveals an intriguing role for TMEM16B in context-independent but not context-dependent fear memory, and supports the notion that dysfunction of the amygdala contributes to anxiety-related behaviors.

DOI: https://doi.org/10.7554/eLife.47106.001

*For correspondence:
Lily.Jan@ucsf.edu

Competing interests: The authors declare that no competing interests exist.

## Introduction

Anxiety disorders are marked by excessive fear. Hence, it is important to study the basic fear circuitry in animal models. In rodent animal models, amygdala, hippocampus, ventromedial hypothalamus, nucleus accumbens, periaqueductal gray and prefrontal cortex have been identified as key brain regions in the neurocircuitry associated with fear responses (*Davis, 2006*; *Maren, 2008*; *Quirk and Mueller, 2008*). Among these brain regions, amygdala play a central role in fear acquisition and expression of fear responses. Investigations of the amygdala microcircuits via optogenetics reveal the importance of inhibitory inputs from central lateral amygdala (CeL) to central medial amygdala (CeM), as well as disinhibition of various output neurons of the central nucleus of the amygdala (CeA), in mediating conditioned fear or anxiolytic processes to reduce anxiety (*Ciocchi et al., 2010*; *Haubensak et al., 2010*; *Janak and Tye, 2015*); acute anxiolytic responses may result from excitatory inputs from basolateral amygdala (BLA) to the CeA, leading to activation of CeL neurons and the ensuing feed-forward inhibition (*Janak and Tye, 2015*; *Paré et al., 2004*; *Tye et al., 2011*). These studies underscore the importance of characterizing neuronal signaling in the amygdala to better understand the physiological basis of fear and anxiety disorders.

*Ano2* encodes an evolutionarily conserved $Ca^{2+}$-activated $Cl^-$ channel (CaCC). As members of the enigmatic TMEM16 (ANO2) family of transmembrane proteins with various physiological functions (*Yang and Jan, 2016*), TMEM16B (ANO2) and TMEM16A (ANO1) are pore forming subunits of the

$Ca^{2+}$-activated $Cl^-$ channels (*Dang et al., 2017*; *Paulino et al., 2017*; *Peters et al., 2015*). *Ano2* mRNA expression is enriched in brain regions including the amygdala, lateral septum, and hippocampus according to the Allen Brain Atlas, raising the intriguing question regarding the role of the TMEM16B $Ca^{2+}$-activated $Cl^-$ channel in anxiety-related behavior. To test whether alteration of TMEM16B activity may contribute to anxiety-related behaviors, we carried out behavioral studies of the knockout mice we generated (*Zhang et al., 2017*). Our finding that *Ano2* knockout mice exhibit impaired anxiety-related behaviors and fear memory indicates that the physiological function of TMEM16B is important for the processing and expression of anxiety.

TMEM16B is known to serve important functions in various brain regions including the hippocampus (*Ha et al., 2016*; *Huang et al., 2012*; *Neureither et al., 2017*; *Wang et al., 2019*; *Zhang et al., 2015*; *Zhang et al., 2017*), however, whether it also modulates neuronal signaling in the amygdala remains an open question. We found that TMEM16B-CaCC is highly enriched in central lateral amygdala (CeL). Loss of TMEM16B function results in action potential broadening in $SOM^+$ CeL neurons that express somatostatin, increase of the amplitude and frequency of GABAergic spontaneous inhibitory postsynaptic current (sIPSC), and reduction of the frequency of miniature IPSC (mIPSC) that depends on presynaptic voltage-gated $Ca^{2+}$ channel activity. Our results provide evidence for a critical role of TMEM16B in neuronal signaling in the amygdala as well as fear and anxiety-like behavior.

## Results

### *Ano2* KO mice display reduced anxiety-related behaviors

To look into the physiological relevance of TMEM16B in anxiety, we performed a battery of anxiety-related behavior tests of 3–4 months old *Ano2* knockout (KO) mice that have sequences for farnesylated mCherry inserted into the coding sequence for *Ano2*, and used their wildtype littermates as controls. First, we subjected each animal to the open field test (OFT) where we could simultaneously record their anxiety-related behaviors and locomotion. As shown in *Figure 1A*, the control (Con) mouse but not the KO mouse tended to avoid the center of the open field. There was a significant increase in the center exploration behavior in *Ano2* KO mice as compared to their control littermates (*Figure 1B*) (unpaired t test in (B), p<0.001, n = 26 for KO and n = 23 for control). This abnormal behavior was not accompanied by any detectable locomotive defects in *Ano2* KO mice, as evident from measurements of the total distance traveled (*Figure 1C,D*).

To further examine anxiety-related behaviors of *Ano2* KO mice, we tested for their performance in the elevated plus maze (EPM). Each individual mouse was given the choice of spending time in the open and unprotected arms or the enclosed and protected arms of the maze. *Figure 1E* shows representative EPM tracks of control and KO mice. Consistent with the OFT results, mice lacking *Ano2* showed significantly greater open-arm exploration (Two-way ANOVA in (F), p=0.0042; Mann Whitney test in (G), p=0.0002 n=26 for KO and n = 23 for control), reflecting a reduction in anxiety-related behaviors as compared to control littermates (*Figure 1F,G*). As in the case of OFT tests, the total distance traveled by *Ano2* KO mice and their control littermates were comparable (*Figure 1H*).

Next, we exposed these mice to a novel environment with protected (dark compartment) and unprotected (light compartment) areas. We found that mice with loss of TMEM16B function entered the light side more frequently and spent more time in this unprotected area as compared to control mice (*Figure 1I,J*) (unpaired t test in (I,J), p<0.001(I), p=0.005 (J), n = 20 for KO and n = 24 for control). Collectively, all these three different assays indicate that *Ano2* KO mice are deficient in anxiety-related behaviors.

Pre-pulse inhibition (PPI) provides a measure of sensorimotor gating that is reduced in several neuropsychiatric disorders. Panic disorder patients have difficulties in progressively dampening their response to sensory and cognitive stimuli, suggesting that they have reduced PPI (*Ludewig et al., 2002*). In our assays, *Ano2* KO mice showed significantly greater pre-pulse inhibition of startle responses as compared with control mice (*Figure 1K*) (Two-way ANOVA in (K), p=0.0370, n = 20 for KO and n = 24 for control). The amplitudes of startle responses induced by stimuli without pre-pulse were comparable between KO and control, as were the control startle response in the absence of any stimuli (*Figure 1L,M*). Thus, *Ano2* KO mice display enhanced PPI and reduced anxiety-related behavior.

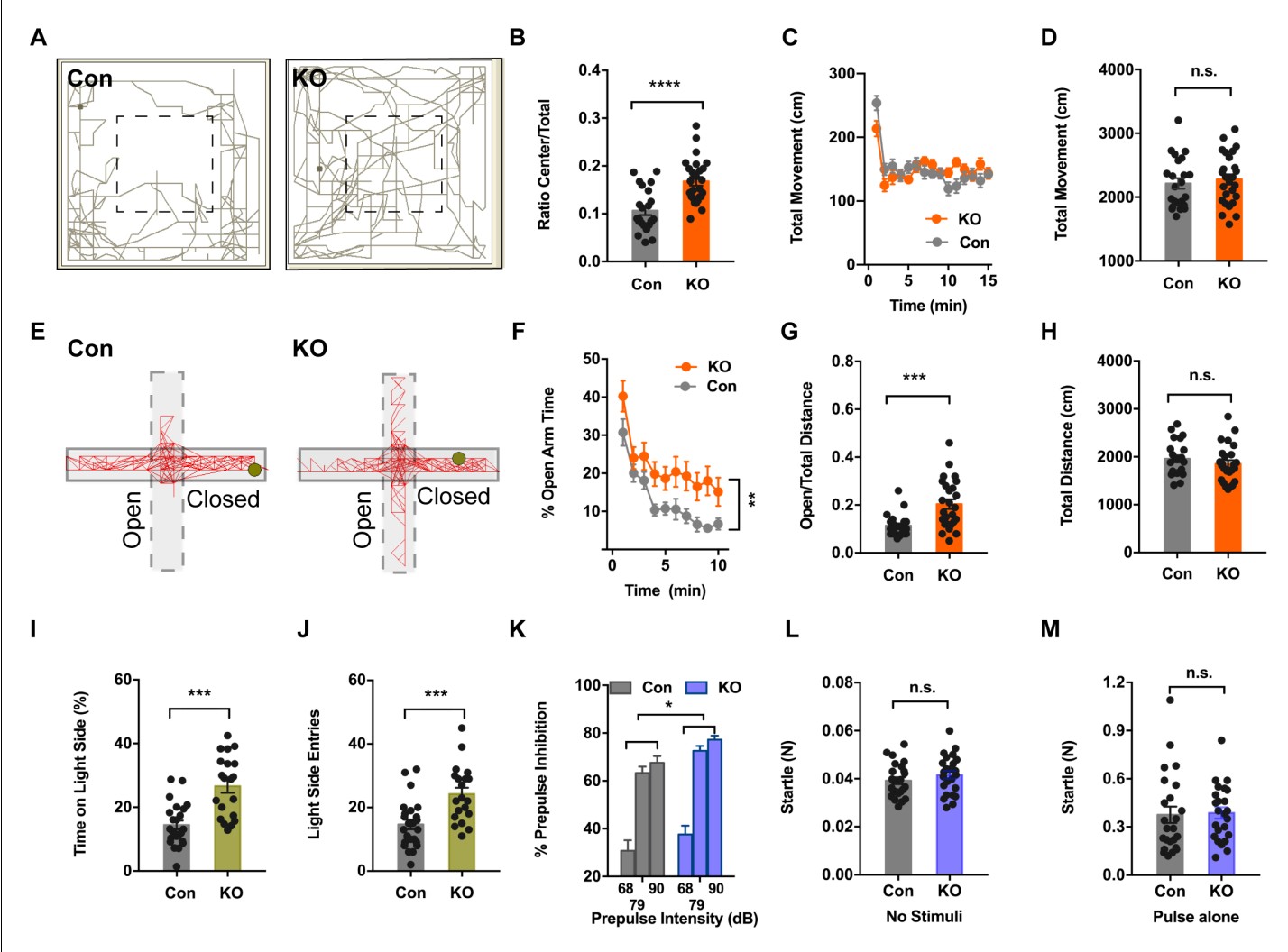

**Figure 1.** *Ano2* Mutant Mice Display Reduced Anxiety-related Behavior and Impaired Pre-pulse Inhibition. (A) Representative traces of *Ano2* KO mice and control (Con) littermates in open field. (B–D) *Ano2* mutant mice display reduced anxiety-related behavior in the open field test, shown as time spent at the center normalized by total time (B). Overall exploratory behavior in the open-field test, shown as distance traveled per minute as function of time (C) and total distance traveled (D), is comparable between mutant and control mice. Two-tail unpaired t test in (B), t = 4.744, df = 46.7, p<0.0001. Two-way ANOVA in (C), F (1,47)=0.04778, p=0.8279. Two-tail unpaired t test in (D), t = 0.5721, df = 47, p=0.5700, n = 26 for KO and n = 23 for control (number of mice tested) for (B–D). (E) Representative elevated plus maze (EPM) tracks of control and *Ano2* KO mice. (F–H) *Ano2* mutant mice display reduced anxiety-related behavior in the EPM test, plotted as time spent in the open arms as function of time (F) and distance traveled in open arms normalized by the total distance traveled (G), with the total distance traveled in the elevated plus maze (H) as control. Two-way ANOVA in (F), F (1, 47)=9.067, p=0.0042. Mann Whitney test in (G), U = 120, p=0.0002. Two-tail unpaired t test in (H), t = 0.9944 df=46, p=0.3252, n = 26 for KO and n = 23 for control (number of mice tested) for F–H. (I and J) *Ano2* mutant mice show reduced anxiety-related behavior in the light-dark box test, shown as the percentage of total time spent in the bright compartment (I) and number of entries to the bright compartments (J) over a 10 min period. Two-tail unpaired t test in (I), t = 4.804, df = 32.94, p<0.0001. Two-tail unpaired t test in (J), t = 3.79, df = 37.77, p=0.0005, n = 20 for KO and n = 24 for control for I and J. (K–M) *Ano2* mutant mice show reduced pre-pulse inhibition (PPI) of startle responses induced by pre-pulses at 68 dB, 79 dB and 90 dB(K). Startle responses to no stimuli (L) and startle response to 120 dB stimuli without pre-pulse (M) are comparable between mutant and control mice. Two-way ANOVA in (K). F (1, 46)=4.612, p=0.0370. Two-tail unpaired t test in (L), t = 1.119, df = 46, p=0.2688. Two-tail unpaired t test in (M), t = 0.1889, df = 46, p=0.8510, n = 24 for KO and n = 24 for control (number of mice tested) for K–M. Data are presented as mean ± SEM. *p<0.05, **p<0.01; ***p<0.001; ****p<0.0001; n.s., no significant difference.

DOI: https://doi.org/10.7554/eLife.47106.002

## *Ano2* knockout mice exhibit altered fear expression

Disturbances in fear learning may contribute to disorders of fear and anxiety, such as panic disorder and specific phobias (*Rosen et al., 1998*). Pavlovian fear conditioning and fear expression has become an important model for investigating the learning and memory processes related to fear in rodents (*Maren, 2001*). We tested for both context-dependent and context-independent fear learning.

Exposing mice to aversive foot shock (Unconditioned Stimuli, US) in an area with contextual stimuli (Conditioned Stimuli, CS) allowed for the testing of context-dependent fear learning and memory. *Ano2* KO and control mice displayed comparable freezing levels during context fear acquisition, which increased as pairings of CS and US were repeatedly presented (*Figure 2A,C*) (Two-way ANOVA, p=0.3928. n = 24 for KO and n = 24 for control). When the same context was presented 24 hr after fear conditioning, *Ano2* KO and control mice exhibited no significant difference in context-dependent fear expression (*Figure 2B,C*) (Two-way ANOVA, p = 0.1472. n = 24 for KO

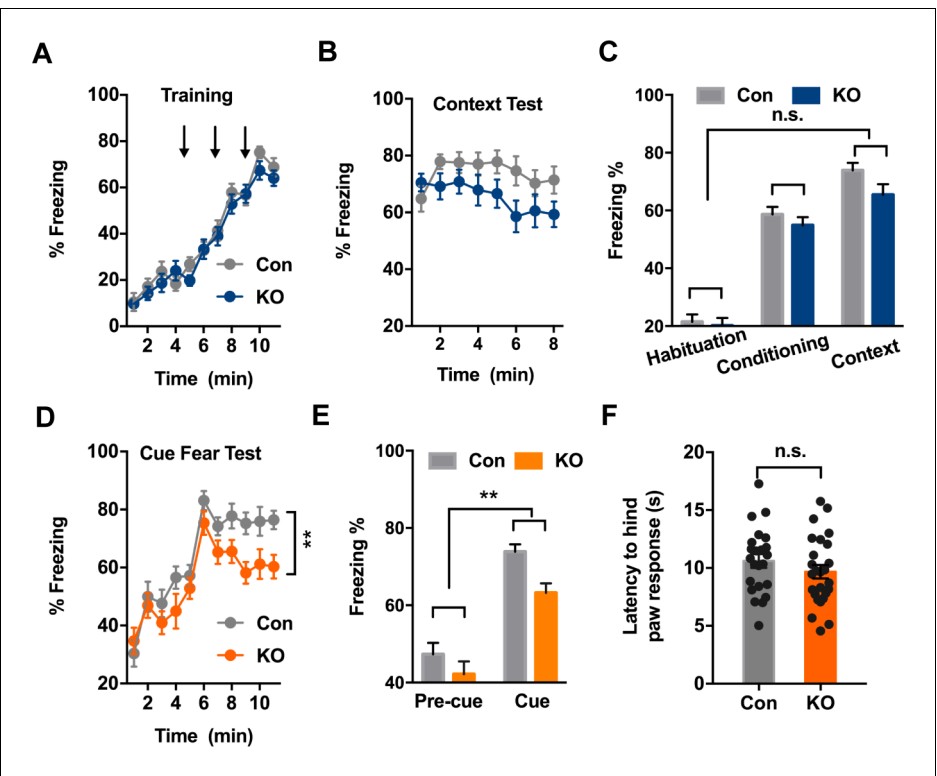

**Figure 2.** *Ano2* Knockout Mice Exhibit Reduced Context-independent Cued Fear Expression. (**A**) *Ano2* KO and control mice exhibited comparable freezing levels during fear acquisition. After habituation for 5 min, freezing levels during fear training with three 30 s 80 dB tones that each was terminated at the same time as a 2 s, 0.45 mA foot shock, separated by 120 s interval for monitoring. Two-way ANOVA in (**A**), F (1, 46)=0.7443, p=0.3928. (**B**) Context-dependent fear recall tested 24 hr after fear conditioning. Mice were placed in the fear conditioning apparatus for 8 min. No shock was presented. Two-way ANOVA in (**B**), F (1, 46)=3.601, p=0.0640. (**C**) *Ano2* KO and control mice exhibited comparable context-dependent fear expression, shown as averaged freezing levels during habituation, conditioning and context test. Two-way ANOVA in (**C**), F (1, 46)=2.173, p=0.1472. (**D and E**) The context-independent cue fear test took place 5–24 hr after the context-dependent test in (**B**). After a 5 min baseline (pre-cue), three 30 s 80 dB tone cues not accompanied with foot shock were presented with 120 s interval for monitoring (cue), in the absence of context cues. Two-way ANOVA in (**D**), F (1, 46)=9.151, p=0.004. Two-way ANOVA in (**E**), F (1, 46)=9.007, p=0.0043. TEMEM16 KO mice displayed lower levels of freezing than control mice during cued fear expression. (**F**) Pain threshold was comparable between control and *Ano2* KO mice.Two tail unpaired t test in (**F**), t = 1.114, df = 46, p=0.2711, n = 24 for KO and n = 24 for control (number of mice tested for A–F). Data are presented as mean ± SEM. *p<0.05, **p<0.01; ***p<0.001; ****p<0.0001; n.s., no significant difference.

DOI: https://doi.org/10.7554/eLife.47106.003

and n = 24 for control), indicating that *Ano2* is not required for context-dependent fear acquisition and expression.

Exposing mice to the aversive foot shock (US) paired with a tone allowed for the testing of context-independent auditory fear learning and memory. *Ano2* KO mice displayed lower levels of freezing than control mice during cued fear expression (*Figure 2D,E*) (Two-way ANOVA, p=0.004, n = 24 for KO and n = 24 for control) while the mutant and control mice displayed similar pain threshold (*Figure 2F*) and similar freezing levels during the acquisition of cued fear learning (*Figure 2A*), highlighting the involvement of TMEM16B for controlling context-independent cued fear expression.

## *Ano2* expression in somatostatin-positive GABAergic neurons in the central lateral amygdala

To examine the cellular expression pattern of *Ano2*, we used RNA scope for its superior sensitivity to detect low abundance transcripts. We found enrichment of *Ano2* mRNA expression in brain regions including the central lateral amygdala, lateral septum and hippocampus, consistent with the expression pattern deposited at the Allen Brain Atlas (*Figure 3A*; *Figure 3—figure supplement 1*). Given that anxiety and fear conditioning involves multiple brain regions including amygdala and hippocampus (*Chen et al., 2016*; *Hall et al., 2001*; *Kochli et al., 2015*; *Vazdarjanova and McGaugh, 1999*; *Weeden et al., 2015*; *Xu et al., 2016*; *Yin et al., 2002*), it is important to examine *Ano2* expression and physiological function in amygdala to assess the potential relevance to anxiety. *Ano2* is mainly expressed in central lateral amygdala (CeL), but not other sub nuclei of amygdala such as basal lateral amygdala (BLA) or the medial part of central amygdala (CeM) (*Figure 3B,C*). CeL, which is composed of several classes of GABAergic inhibitory neurons (*Cassell and Gray, 1989*; *Ehrlich et al., 2009*; *Haubensak et al., 2010*), modulates conditioned fear response and anxiety-related behavior via inhibition of neurons in the central lateral and medial subdivisions of CeA, which provide the major output of amygdala (*Ciocchi et al., 2010*; *Janak and Tye, 2015*; *LeDoux et al., 1988*; *Tye et al., 2011*). GABAergic inhibitory neurons in CeL can be classified by expressions of distinct neurochemical markers (*Ehrlich et al., 2009*; *Haubensak et al., 2010*). Among these subtypes, somatostatin-positive (SOM$^+$/*Sst*) neurons constitute a major population, and are intermingled with protein kinase C-δ expressing (PKC-δ$^+$/*Prkcd*) neurons and neurons containing corticotropin-releasing hormone (*Crh*), neurotensin (*Nts*) and tachykinin 2 (*Tac2*); neurons expressing various combinations of these markers account for the majority (96%) of the GABAergic neurons in CeL (*Kim et al., 2017*). Of these markers, *Prkcd* and *Sst* almost have no overlap with each other (2%–13%) (*Kim et al., 2017*; *Li et al., 2013*), while *Sst* exhibits high levels of overlap (>50%) with *Crh$^+$/Nts$^+$/Tac2$^+$*. Our RNA scope study revealed that the majority (92.6%) of SOM$^+$ (*Sst*) neurons expressed *Ano2*, and the majority (71.4%) of *Ano2* positive cells displayed *Sst* immunoreactivity (*Figure 3E*). We further confirmed that SOM$^+$ (*Sst*) neurons expressed glutamate decarboxylase 2 (*Gad2*), a marker of inhibitory neurons (*Figure 3D*).

## TMEM16B-mediated Ca2+-activated Cl$^−$ current in somatostatin-positive neurons of central lateral amygdala

For studies of TMEM16B-mediated Ca$^{2+}$-activated Cl$^-$ current (CaCC) in SOM$^+$ cells, we used a *Sst-IRES-Cre* knock-in mouse line, with *Cre* driven by the endogenous promoter of the *Somatostatin* (*Sst*) gene (*Taniguchi et al., 2011*). SOM$^+$ neurons in *Sst-IRES-Cre;*Ai14 mice can be readily identified by the red fluorescence of the tdTomato reporter of Ai14 (*Figure 4A*).

Given that TMEM16B-CaCC is expressed in the hippocampus, lateral septum, and Inferior olive neurons, where it is activated by Ca$^{2+}$ influx through voltage-gated Ca$^{2+}$ channels (*Huang et al., 2012*; *Wang et al., 2019*; *Zhang et al., 2017*), we tested whether TMEM16B can also be activated by Ca$^{2+}$ influx through voltage-gated Ca$^{2+}$ channels in neurons of central lateral amygdala. First, we confirmed that Ca$^{2+}$ currents were elicited in SOM$^+$ neurons with depolarizations lasting 60 ms (*Figure 4—figure supplement 1A*). Next, we carried out whole cell recording in the presence of 5 mM QX314, 110 mM Cs$^+$, and 20 mM tetraethylammonium (TEA), to block voltage-gated Na$^+$ channels and the majority of K$^+$ channels including Ca$^{2+}$-activated K$^+$ channels. We found that depolarization to +10 mV for 500 ms elicited a transient inward current as a result of the superimposition of an inward Ca$^{2+}$ current and an outward CaCC that developed over time leading to the appearance of a

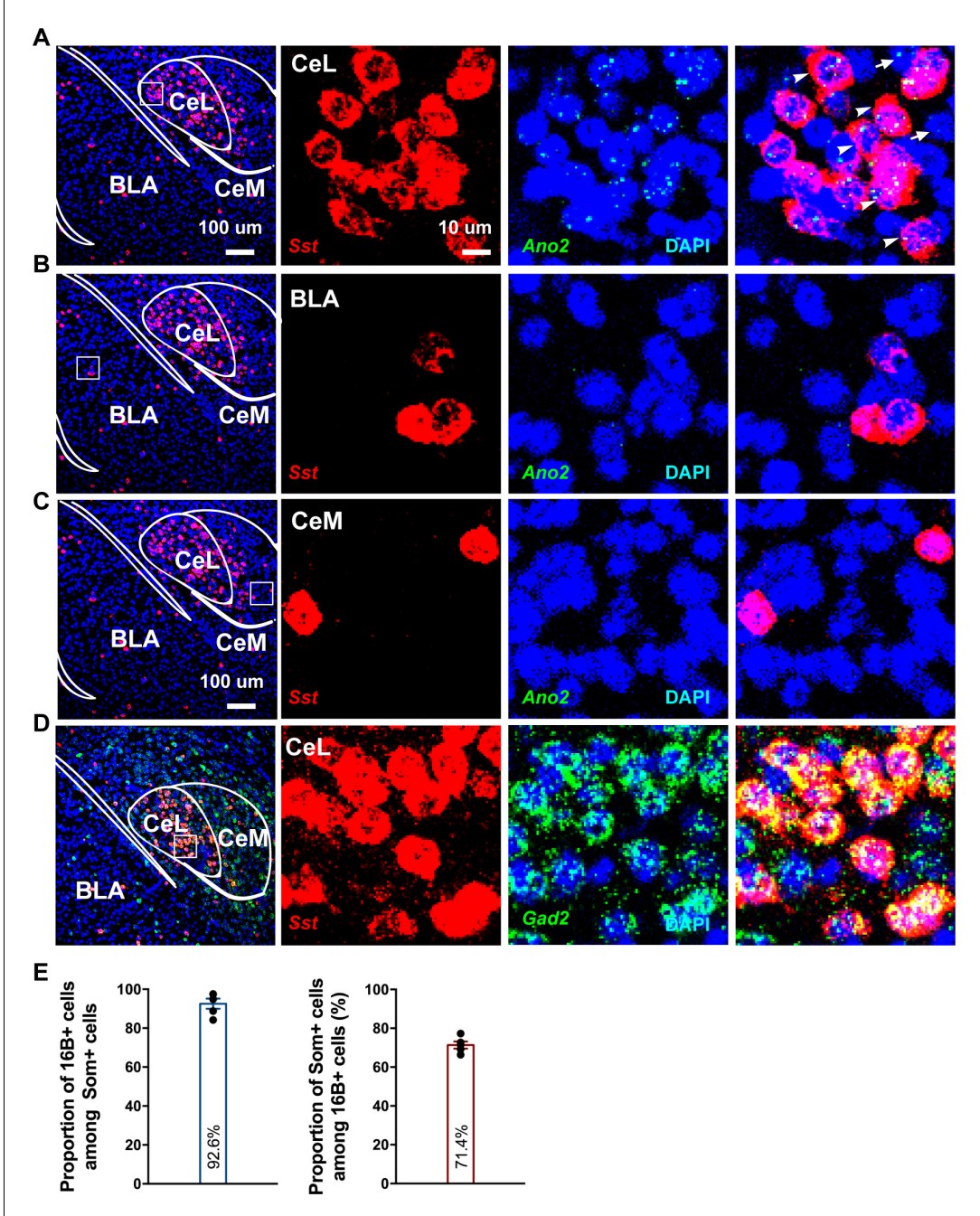

Figure 3. *Ano2* mRNA Expression in Somatostatin-positive GABAergic Neurons in the Central Lateral Amygdala. (A–C) RNA scope studies reveal that *Ano2* is expressed in central lateral amygdala (CeL), but not in basal lateral amygdala (BLA) or central medial amygdala (CeM). Representative images of immunofluorescence staining for *Sst* (red) and RNA scope labeling of *Ano2* mRNA (green). Cell nuclei are stained with DAPI (blue). (D) Representative images of double immunofluorescence staining for *Gad2* (green) and *Sst* (red). Cell nuclei are stained with DAPI (blue). (E) Left, percentage of *Sst*-positive neurons that express *Ano2* mRNA. Right, percentage of *Ano2* mRNA-positive neurons that express *Sst*. n = 653 neurons from five mice.

DOI: https://doi.org/10.7554/eLife.47106.004

The following figure supplement is available for figure 3:

**Figure supplement 1.** *Ano2* mRNA Expression Patterns.

DOI: https://doi.org/10.7554/eLife.47106.005

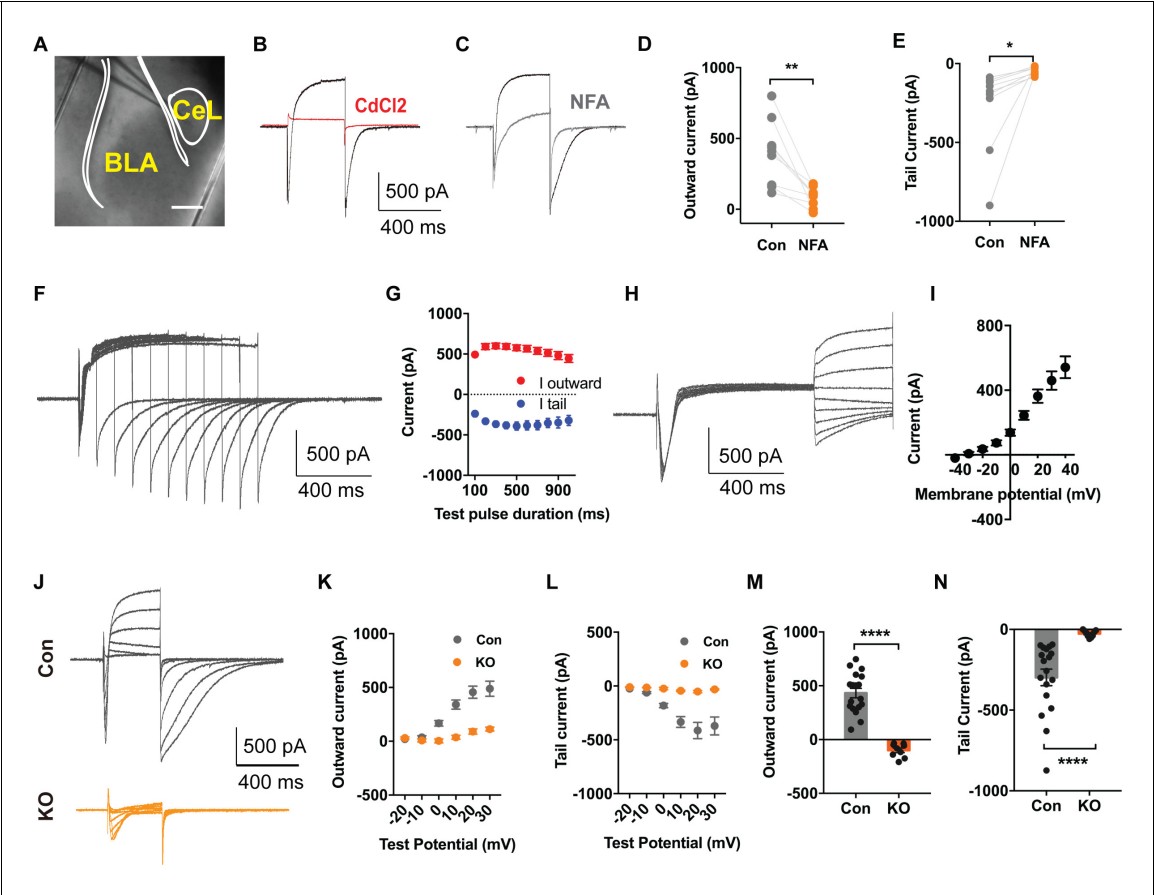

**Figure 4.** TMEM16B Mediated Ca$^{2+}$-activated Cl$^-$ Currents in Somatostatin-positive Neurons of the Central Lateral Amygdala. (**A**) Brain slice of *Sst-IRES-Cre* and Ai14 reporter mice with fluorescently marked somatostatin-positive (SOM$^+$) neurons in the amygdala. Scale bar, 200 μm. (**B and C**) Depolarization of SOM$^+$ neurons to +10 mV elicited a Ca$^{2+}$ current superimposed with a large outward current that developed over time, followed with a tail current upon hyperpolarization to −80 mV. The outward current and tail current can be inhibited by 100 μM niflumic acid (NFA) (**C**), and it is completely abolished by 200 μM CdCl$_2$, a blocker of voltage-gated Ca$^{2+}$ channels, indicating that it is induced by Ca$^{2+}$ influx (**B**). (**D and E**) NFA reduced the outward current and tail current. Paired t test in (**D**), t = 4.392, df = 8, p=0.0023. Paired t test in (**E**), t = 2.55, df = 8, p=0.0342, n = 9 cells for each group from three mice per group for D and E. (**F and G**) Summary plot of the peak amplitude of the outward current and tail current as a function of the duration of the test pulse to +10 mV. n = 7 cells from three mice. (**H and I**) Measurements of reversal potential indicating that the current induced by Ca$^{2+}$ influx in SOM$^+$ neurons arises from Ca$^{2+}$-activated Cl$^-$ channels (CaCCs). Membrane potential was depolarized to +10 mV to activate Ca$^{2+}$ channels and then held from −40 mV to +40 mV in 10 mV increments for reversal potential measurement, followed with repolarization to −80 mV. n = 15 cells from four mice. (**J–L**) Genetic ablation of *Ano2* completely eliminated the outward current and tail current in SOM$^+$ neurons, indicating that TMEM16B CaCC mediates the Ca$^{2+}$-sensitive outward current and tail current in SOM$^+$ neurons. Summary plot of the outward current (**K**) and the tail current (**L**) at various membrane potentials. Membrane potential was depolarized from −20 mV to +30 mV in 10 mV increments to activate Ca$^{2+}$ channels, and then repolarized to −80 mV. (**M and N**) Comparison of the amplitudes of the outward current and tail current of SOM+ neurons from *Ano2* KO mice and control mice. Mann Whitney test in (**M**), U = 0, p<0.0001. Mann Whitney test in (**N**), U = 0, p<0.0001. Cells were held at −80 mV. n = 18 cells from 5 KO and n = 9 cells from four control for K, L, M and N. Data are presented as mean ± SEM. *p<0.05; **p<0.01; ***p<0.001; ****p<0.0001; n.s., no significant difference.

DOI: https://doi.org/10.7554/eLife.47106.006

The following figure supplement is available for figure 4:

**Figure supplement 1.** No detectable CaCC in non-SOM+ neurons.

DOI: https://doi.org/10.7554/eLife.47106.007

net outward current following the transient inward current, followed by a slowly deactivating tail CaCC current upon repolarization to −80 mV (*Figure 4B*). The feature of CaCC we recorded is similar to that reported previously (*Zhang et al., 2017*). Treatment of brain slices with CdCl$_2$, a Ca$^{2+}$ channel blocker, abolished both the inward Ca$^{2+}$ current and the CaCC (*Figure 4B*), indicating that CaCC activation is triggered by Ca$^{2+}$ influx through voltage-gated Ca$^{2+}$ channels. To verify that the

outward current superimposed on the inward $Ca^{2+}$ current during depolarization and the tail current upon repolarization are $Cl^-$ currents, we first showed that NFA, a $Cl^-$ channel blocker, reduced both the outward current and the tail current (*Figure 4C,D,E*) (Paired t test in (*D*), p=0.0023; Paired t test in (*E*), p=0.0342, n = 9 cells for each group from three mice per group). For further validation, we varied the duration of the $Ca^{2+}$ channel activation and hence the extent of $Ca^{2+}$ entry. The peak amplitudes of the outward current and the tail current increased and reached a plateau with prolonged $Ca^{2+}$ channel activation (*Figure 4F,G*). We further measured the reversal potential of the tail current and found that it corresponds to the $Cl^-$ equilibrium potential (*Figure 4H,I*). Moreover, no obvious CaCC current is elicited in non-SOM+ neurons (*Figure 4—figure supplement 1B,C,D,E*). These findings support the notion that the $Ca^{2+}$-sensitive outward current in SOM+ neurons is mediated by $Ca^{2+}$-activated $Cl^-$ channels (CaCCs).

Finally, we recorded from brain slices of *Sst-IRES-Cre*;Ai14 mice lacking TMEM16B to determine whether the outward current is eliminated in *Ano2* KO mice. We held the SOM+ neurons at different membrane potentials and measured the outward current (*Figure 4J,K*) and tail current (*Figure 4J, L*). In contrast to control brain slices, depolarization of SOM+ neurons lacking TMEM16B elicited a fast inward calcium current without the outward current and tail current (*Figure 4M,N*) (Mann Whitney test in (*M, N*), p<0.0001(M,N), n = 18 cells from 5 KO and n = 9 cells from four control for *K, L, M* and *N*), indicating that TMEM16B is solely responsible for CaCC in those GABAergic neurons.

## TMEM16B-CaCC regulates action potential waveform of somatostatin-positive neurons in central lateral amygdala

To investigate whether TMEM16B-mediated CaCC activity regulates the physiological property of SOM+ cells, we first performed whole-cell current-clamp recordings with acute CeL slices from control *Sst-IRES-Cre*;Ai14 mice and *Sst-IRES-Cre;Ano2-/-*;Ai14 mutants. We assessed neuronal excitability by applying a series of current injections each lasting 500 ms. Whereas *Ano2* KO and control littermates had comparable numbers of action potentials generated, firing frequency adaptation, threshold, spike amplitude, after hyperpolarization (AHP), resting membrane potential (RMP), and passive membrane properties (*Figure 5A,B,C,G*; *Figure 5—figure supplement 1*), there was a significant increase in the half width, half width adaptation, and the width of the action potential at 90% from the peak (APD90%) in *Ano2* KO mice as compared with control littermates (*Figure 5A,D, E,F*) (Mann Whitney u test in (*D, E, F*), p=0.0145 (*D*), p=0.0091 (*E*), p=0.0003 (*F*), n = 13 cells for KO and 14 cells for control, from three mice each).

## Altered inhibitory neurotransmission in central lateral amygdala of *Ano2* mutant mice

Abnormalities in GABAergic neurotransmission are associated with the neurobiology of anxiety (*Makkar et al., 2010*; *Sanders and Shekhar, 1995*). Central to the circuits of fear and anxiety, local inhibition in central lateral amygdala (CeL) plays a crucial role in the processing of emotions (*Ciocchi et al., 2010*; *Tye et al., 2011*). To date, however, little is known about the molecular mechanisms of inhibitory synapses within CeL that may contribute to the pathophysiology of anxiety disorders. As TMEM16B-CaCC is largely present in SOM+ GABAergic inhibitory neurons in CeL, we wondered whether TMEM16B-CaCC may affect GABAergic neurotransmission in CeL, which is involved in fear and anxiety-related behaviors (*Janak and Tye, 2015*).

Previous studies employing optogenetics have shown that CeL SOM+ and SOM- neurons mutually inhibit one another (*Li et al., 2013*). Likewise, CeL PKC-δ+ and PKC-δ- neurons innervate one another (*Haubensak et al., 2010*). In addition, CeL neurons are highly interconnected, and the majority of common local synaptic connections are between SOM+ neurons, as revealed with paired recording (*Hunt et al., 2017*). Whereas CeL PKC-δ+ interneurons project to CeM (*Haubensak et al., 2010*), CeL SOM+ neurons provide potent inhibition within CeL rather than direct inhibition of CeM neurons (*Li et al., 2013*). To determine whether inhibitory transmission is altered in *Ano2* KO mice, we measured the spontaneously occurring action potential dependent inhibitory postsynaptic currents (sIPSCs) and action potential independent miniature inhibitory postsynaptic currents (mIPSCs) of SOM+ cells in CeL slices from *Ano2* KO mice and control littermates. Given that the waveform of action potential controls synaptic efficacy in pyramidal neurons (*Kole et al., 2007*), it is possible that the action potential broadening in *Ano2* KO mice could lead to an increase in transmitter release.

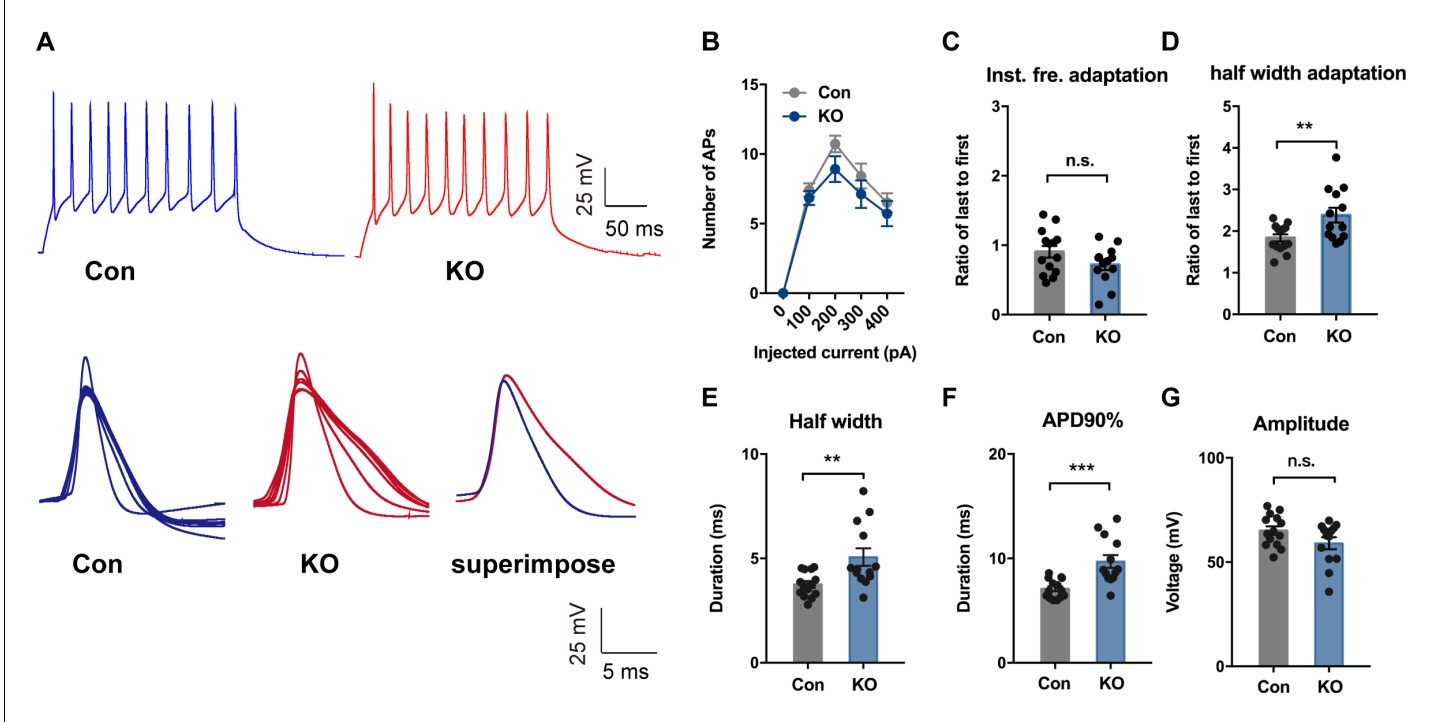

**Figure 5.** TMEM16B-CaCC Regulates Action Potential Waveform of Somatostatin-positive Neurons in the Central Lateral Amygdala. (A) upper panel, representative traces of action potentials in SOM+ neurons from control and *Ano2* KO mice in response to 100 pA current pulse lasting for 500 ms. Lower panel, overlay of action potentials induced by the current pulse of neurons with (left) or without (middle) *Ano2*, with superimposition of the averaged spikes from neurons with or without *Ano2* shown on the right. (B) Number of spikes induced by step current injection in SOM+ neurons from control and *Ano2* KO mice. Two-way ANOVA in (B), $F_{(1, 49)}=1.804$, $p=0.1855$. n = 25 cells for KO and 26 cells for control, from seven mice each. (C) Ratio of the last to the first instantaneous firing frequency, from control and *Ano2* KO brain slices. Two tail Unpaired t test, $t = 1.635$, $df = 25$, $p=0.1146$. n = 13 cells for KO and 14 cells for control, from three mice each. (D) Ratio of the last to the first half width, from control and *Ano2* KO brain slices. Mann Whitney u test in (D), $U = 41$, $p=0.0145$. n = 13 cells for KO and 14 cells for control, from three mice each. (E) Half width of action potentials in SOM+ neurons with or without *Ano2*. Mann Whitney test, $U = 38$, $p=0.0091$. n = 13 cells for KO and 14 cells for control, from three mice each. (F) The averaged action potential duration at 90% repolarization (APD90%) in SOM+ neurons from control and *Ano2* KO mice. Mann Whitney u test in (F), $U = 21$, $p=0.0003$. n = 13 cells for KO and 14 cells for control, from three mice each. (G) Average amplitude of action potential in SOM+ neurons with or without *Ano2*. Two tail Unpaired t test, $t = 1.771$, $df = 25$, $p=0.0888$. n = 13 cells for KO and 14 cells for control, from three mice each. Data are presented as mean ± SEM. *$p<0.05$, **$p<0.01$; ***$p<0.001$; ****$p<0.0001$; n.s., no significant difference.

DOI: https://doi.org/10.7554/eLife.47106.008

The following figure supplement is available for figure 5:

**Figure supplement 1.** SOM+ Neurons Lacking *Ano2* Have Normal Resting Potential, Membrane Capacitance, Input Resistance, Time Constant, Voltage threshold and Afterhyperpolarization.

DOI: https://doi.org/10.7554/eLife.47106.009

Indeed, loss of TMEM16B function led to an increase of sIPSC amplitude and frequency (*Figure 6*) (Mann Whitney test in (B), $p=0.0004$ for amplitude; $p=0.0021$ for frequency. n = 31 cells for KO and n = 29). As both the amplitude and frequency of sIPSC were increased in KO neurons, we considered two possible underlying mechanisms. First, the effect may reflect increased presynaptic transmitter release. Second, It may include both increased presynaptic transmitter release and alteration of the postsynaptic response. To distinguish between these two possibilities, we evaluated the GABA receptor function in control and *Ano2* KO SOM+ neurons, by puffing 100 μm GABA (50 ms, 4–6 psi) to generate evoked inhibitory postsynaptic currents (eIPSC). This eIPSC can be eliminated by applying PTX in the bath solution along with GABA puffing (*Figure 6—figure supplement 1A*). The amplitude of eIPSC was comparable between control and TMEM16B knockout SOM+ neurons (*Figure 6—figure supplement 1B*) (Unpaired t test, $p=0.2628$. n = 9 cells for each genotype from three mice each), indicating that TMEM16B activity does not alter GABA receptor function. Our findings thus support the notion that the potentiation of sIPSC could have arisen from the broadening of

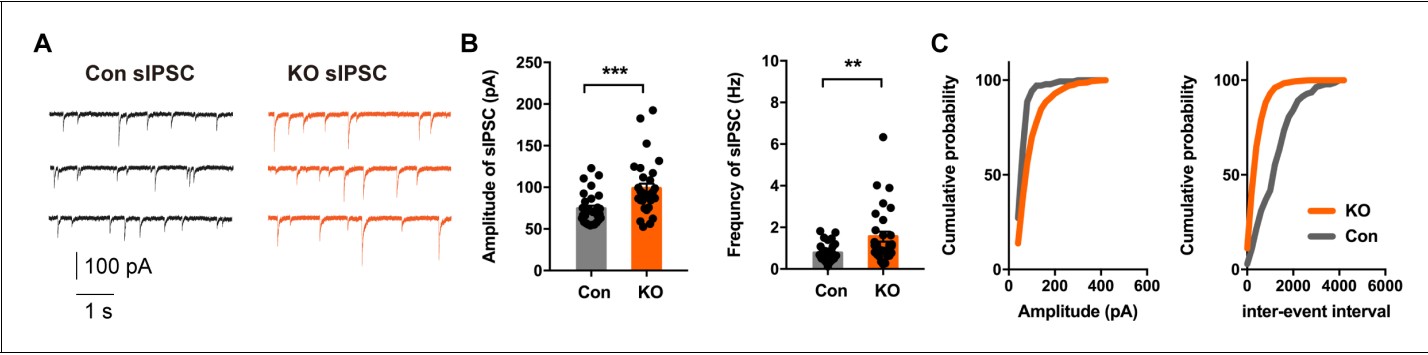

**Figure 6.** Altered Inhibitory Neurotransmission in the Central Lateral Amygdala of *Ano2* KO mice. (**A**) Representative traces of sIPSCs from SOM[+] neurons of *Ano2* KO and control mice. (**B**) The SOM[+] neurons of *Ano2* KO mice showed increased sIPSC frequency and sIPSC amplitude. Mann Whitney test in (**B**), U = 215, p=0.0004 for amplitude of sIPSC. U = 245, p=0.0021 for frequency of sIPSC. n = 31 cells for KO and n = 29 cells for control, from 10 mice each. (**C**) Representative cumulative probability plot of sIPSC frequency and amplitude of SOM[+] neurons in CeL from *Ano2* KO and control mice. Data are presented as mean ± SEM. *p<0.05, **p<0.01; ***p<0.001; ****p<0.0001; n.s., no significant difference.
DOI: https://doi.org/10.7554/eLife.47106.010

The following figure supplement is available for figure 6:

**Figure supplement 1.** Similar eIPSC amplitudes of SOM+ neurons in control and *Ano2* knockout mice.
DOI: https://doi.org/10.7554/eLife.47106.011

action potentials in presynaptic neurons lacking TMEM16B. These studies reveal that the inhibitory neurotransmission is abnormal in the CeL of *Ano2* KO mice, as evident from the potentiation of the sIPSC due to spontaneous network activities of the CeL.

## Voltage-gated Ca$^{2+}$ channels are involved in the presynaptic regulation of spontaneous transmitter release by TMEM16B-CaCC

Not only do voltage-gated Ca$^{2+}$ channels mediate Ca$^{2+}$ influx to trigger transmitter release that is evoked by an action potential, Ca$^{2+}$ channels have also been found to be physiological triggers for spontaneous transmitter release at mouse inhibitory synapses (*Williams et al., 2012*). By blocking action potential generation with the Na$^+$ channel blocker TTX, we examined spontaneous release of GABA as revealed by mIPSC recordings. Whereas the amplitude of mIPSC appeared normal in CeL slices of *Ano2* KO mice, the frequency of mIPSC was significantly reduced (*Figure 7A,B*) (Mann Whitney test in (*B, left panel*), p = 0.0016; unpaired t test in (*B, right panel*), p = 0.2283, n = 16 cells for KO and n = 15 cells for control, from 5 mice each). As revealed by the plot of the cumulative probability of inter-event interval and mIPSC amplitude in *Figure 7C*, the frequency of spontaneous transmitter release was reduced in the *Ano2* KO mice. Consistent with a previous study (*Williams et al., 2012*), application of Cd$^{2+}$ (200 μM), a Ca$^{2+}$ channel blocker, reduced the mIPSC frequency by 54.74% ± 4.032% without affecting the mIPSC amplitude (*Figure 7D-H*), thus implicating Ca$^{2+}$ channels in spontaneous GABA release in central lateral amygdala. On average, about 46% of the mIPSCs were Cd$^{2+}$ resistant, suggesting that other mechanisms also contribute to the regulation of spontaneous GABA release. CdCl$_2$ had a much smaller effect on the mIPSC frequency in KO mice (22.52% ± 6.326%, *Figure 7F, H*), while the frequency of CdCl$_2$ insensitive mIPSC was comparable between control and KO (*Figure 7F*). These findings provide evidence for TMEM16B function in the nerve terminal, including regulation of spontaneous GABA release that depends on Ca$^{2+}$ channel activity.

## Discussion

In this study, we report that loss of TMEM16B (*Ano2*) function leads to behavioral abnormalities, including impaired anxiety-related behaviors and context-independent auditory fear expression. At the cellular level, TMEM16B constitutes a previously unknown Ca$^{2+}$-activated Cl$^-$ channel in CeL GABAergic neurons with somatostatin expression. In addition, we found that SOM$^+$ neurons in

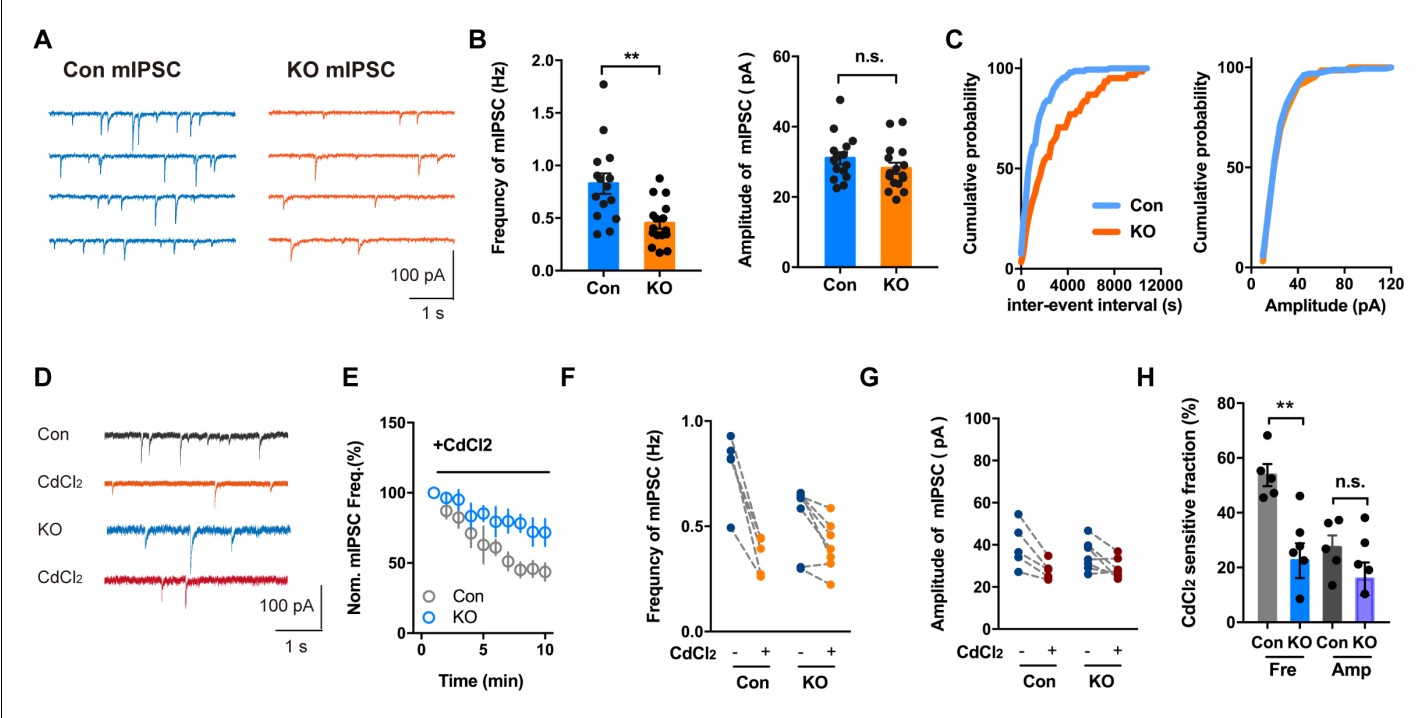

**Figure 7.** Voltage-activated $Ca^{2+}$ Channel Involvement in mIPSC in Brain Slices With or Without *Ano2*. (A) Representative traces of mIPSCs from SOM[+] neurons of *Ano2* KO and control mice. (B) The SOM+ neurons of *Ano2* KO mice showed a decrease of mIPSC frequency, and normal mIPSC amplitude. Mann Whitney test in (B, left panel), U = 42.5, p=0.0016 for frequency of mIPSC. Two tail unpaired t test in (B, right panel), t = 1.231, df = 29, p=0.2283 for amplitude of mIPSC. n = 16 cells for KO and n = 15 cells for control, from five mice each. (C) Representative cumulative probability plot of mIPSC frequency and amplitude of SOM[+] neurons in CeL from *Ano2* KO and control mice. (D) Representative traces of mIPSCs from *Ano2* KO and control brain slices in basal solution with or without $Cd^{2+}$. (E) $Cd^{2+}$ treatment reduced the mIPSC frequency of SOM[+] neurons in the central lateral amygdala from KO mice and control littermates, as shown in the plot of normalized mIPSC frequency as function of time. Data are normalized to the baseline of mIPSC recorded in the first 3 min before $Cd^{2+}$ application. (F) The effects of $Cd^{2+}$ treatment on the mIPSC frequency in SOM[+] neurons from control and KO mice. (G) The effects of $Cd^{2+}$ treatment on the mIPSC amplitude in SOM[+] neurons from control and KO mice. (H) The $Cd^{2+}$-sensitive fraction of mIPSCs in SOM+ neurons from central lateral amygdala of KO mice and control littermates. Two tail Unpaired t test, t = 3.765, df = 10, p=0.0037 for $Cd^{2+}$ sensitive mIPSC frequency. t = 1.424, df = 10, p=0.1848 for $Cd^{2+}$ sensitive mIPSC amplitude. n = 7 cells from KO and n = 5 cells from control, three mice each. Data are presented as mean ± SEM. *p<0.05, **p<0.01; ***p<0.001; ****p<0.0001; n.s., no significant difference.

DOI: https://doi.org/10.7554/eLife.47106.012

central lateral amygdala of *Ano2* mutant mice have broadened action potential. The lengthening of action potential duration is associated with potentiation of the action potential evoked inhibitory postsynaptic current (sIPSC). Our finding that ablation of TMEM16B also causes a decrease of the mIPSC frequency provides further evidence for the functional significance of TMEM16B in the presynaptic nerve terminal. Intriguingly, we found that TMEM16B regulates miniature inhibitory postsynaptic current (mIPSC) by modulating the frequency of spontaneous GABA release that depends on voltage-gated $Ca^{2+}$ channel activity. In summary, our results provide evidence for a critical role of TMEM16B in the regulation of SOM[+] GABAergic neuronal signaling and inhibitory transmission in central lateral amygdala, as well as the involvement of TMEM16B in fear and anxiety-like behaviors.

Otowa and colleagues conducted GWAS studies of panic disorder (PD) in the Japanese population and found seven SNPs that are significantly associated with PD, which are located in or adjacent to genes such as *PKP1*, *PLEKHG1*, *TMEM16B*, *CALCOCO1*, *SDK2* and *CLU* (*Otowa et al., 2009*). However, two follow up studies from the same group failed to show any significant association between SNPs in *TMEM16B* and PD (*Otowa et al., 2012*; *Otowa et al., 2010*). Our finding that loss of TMEM16B function results in a reduction of context-independent fear memory and anxiety-related behaviors in mice allows us to look into the physiological functions of TMEM16B of potential relevance to anxiety and panic disorders.

TMEM16B expression is mainly confined to the nervous system, including the olfactory neurons (*Billig et al., 2011*; *Pietra et al., 2016*), retina (*Mercer et al., 2011*; *Stöhr et al., 2009*), hippocampus (*Huang et al., 2012*), lateral septum (*Wang et al., 2019*), and inferior olive neurons (*Zhang et al., 2017*). In the hippocampus, lateral septum and inferior olive, TMEM16B modulates action potentials and neuronal excitability. In the photoreceptor terminals of mouse retina, a region where TMEM16B forms a complex with the anchor protein PSD95, TMEM16B may regulate glutamate release. Our study focused on the physiological functions of TMEM16B in the amygdala, the site for the formation and storage of fear memories crucial for anxiety (*Davis, 1992*; *LeDoux et al., 1988*). Alterations in the structure (*Hayano et al., 2009*) and activity (*Pfleiderer et al., 2007*; *Shin and Liberzon, 2010*) of the amygdala have been reported in patients with anxiety and panic disorder (PD), indicating that the amygdala has a pivotal role in the pathogenesis of these diseases. Our finding that *Ano2* KO mice displayed impaired auditory fear expression, raises the intriguing possibility that TMEM16B expressing neurons in the amygdala contribute to the regulation of fear and anxiety-related behaviors. It remains possible that TMEM16B-CaCC in other brain areas may also contribute to fear and anxiety-like behaviors. In fact, ventral hippocampus is involved in defensive fear related behavior which is revealed by elevated plus maze (*Kjelstrup et al., 2002*). It would be of interest to conduct a more targeted knockout approach and test how removal *Ano2* from SOM$^+$ neurons of CeL affects anxiety-related behaviors. Ablation and pharmacological experiments implicate both hippocampus and amygdala in context-dependent and context-independent fear conditioning (*Chen et al., 2016*; *Hall et al., 2001*; *Kochli et al., 2015*; *Vazdarjanova and McGaugh, 1999*; *Weeden et al., 2015*; *Xu et al., 2016*; *Yin et al., 2002*). Intriguingly, loss of TMEM16B function affected context-independent but not context-dependent fear memory, indicating that different forms of fear memory may involve circuitry with different molecular features of neuronal regulation.

GABA agonists and antagonists can modulate fear (*Makkar et al., 2010*) and anxiety-like behaviors (*Barbalho et al., 2009*; *Sanders and Shekhar, 1995*). Central lateral amygdala has emerged as a primary site for the diazepam anxiolytic action (*Griessner et al., 2018*). The diazepam-binding GABA$_A$-receptor subunits (*Olsen and Sieghart, 2009*) are expressed at higher levels in SOM$^+$ neurons than in SOM$^-$ neurons. In addition, application of diazepam increases both the frequency and the size of spontaneous inhibitory postsynaptic currents (sIPSCs) of SOM$^+$/PKCδ$^-$ neurons but not SOM$^-$/PKCδ$^+$ neurons (*Griessner et al., 2018*). Moreover, we found that elimination of TMEM16B-CaCC resulted in prolonged action potentials and increased inhibitory postsynaptic current in SOM$^+$ neurons of the central lateral amygdala, reminiscent of the effect of diazepam on SOM$^+$ neurons. This alteration in neuronal signaling may alter the inhibitory tone in the central amygdala (CeA), thereby affecting disinhibition of CeA output neurons that are involved in anxiety and conditioned fear memory in *Ano2* KO mice. For example, increased synaptic inhibition of SOM$^+$ CeL neurons could dampen the excitatory input from BLA to SOM$^+$ neurons that drives fear expression (*Figure 8*) (*Janak and Tye, 2015*).

SOM$^+$ CeL neurons receive excitatory inputs from BLA neurons that display learning-related plasticity during fear conditioning (*Li et al., 2013*). These SOM$^+$ neurons project to the periaqueductal gray (PAG) and the paraventricular nucleus of the thalamus (PVT) and form part of the circuit for fear (*Janak and Tye, 2015*; *Penzo et al., 2014*). TMEM16B-CaCC modulates inhibition of SOM$^+$ CeL neurons, as evident from the increased sIPSCs (*Figure 6*) associated with action potential broadening that is caused by loss of TMEM16B function (*Figure 8A*). Increased inhibition of SOM$^+$ CeL neurons in *Ano2* KO mice could reduce the extent of their excitation by BLA neurons as well as their ability to inhibit downstream neurons in PAG, thus resulting in a reduction of fear and anxiety-related behaviors (*Figure 8B*). This scenario is in accordance with the reduced auditory fear expression and anxiety-like behaviors of *Ano2* KO mice. The correlation between alteration in the SOM$^+$ neuronal activity and GABAergic transmission and the impaired fear and anxiety-like behaviors in *Ano2* KO mice is consistent with the notion that inhibitory transmission in CeL plays a key role in the modulation of fear expression and anxiety-like behaviors.

CaCC can be activated by high threshold Cav channel in inferior olive neurons (*Zhang et al., 2017*). In our case, SOM$^+$ neurons exhibited CaCC when the membrane potential was depolarized beyond −10 mV (*Figure 4J,K,L*), indicating that TMEM16B-CaCC activation is triggered by Ca$^{2+}$ influx through dendritic high-threshold Ca$_V$ channels. The R type Ca$^{2+}$ channel is a major source of dendritic Ca$^{2+}$ influx in response to action potentials (*Magee and Johnston, 1995*; *Sabatini and Svoboda, 2000*). Interestingly, the R type Ca$^{2+}$ channel is highly enriched in central amygdala

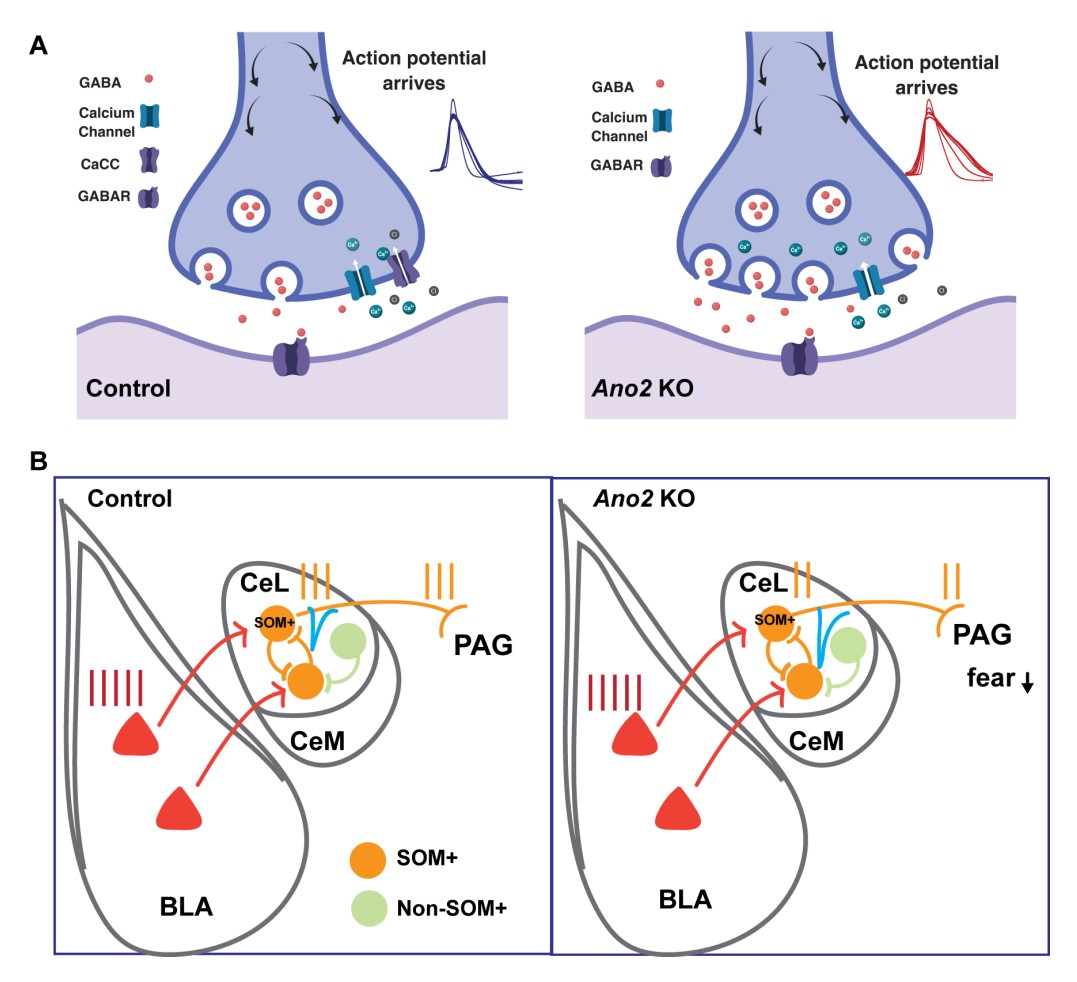

**Figure 8.** Mechanistic model for how presynaptic loss of function of TMEM16B, a $Ca^{2+}$-activated $Cl^-$ channel, in central lateral amygdala (CeL) $SOM^+$ GABAergic neurons mediates the electrophysiological and behavioral phenotypes. (**A**) Loss of TMEM16B mediated CaCC in $SOM^+$ CeL neurons results in broadening of action potential, thereby increasing presynaptic calcium channel activity and GABA release evoked by spike firing. (**B**) Increased inhibition tone onto $SOM^+$ CeL neurons could contribute to the reduction of fear and anxiety-related behaviors in *Ano2* KO mice, by reducing the extent that $SOM^+$ CeL neurons are excited by neurons in the basolateral complex of the amygdala (BLA) as well as reducing the extent that $SOM^+$ CeL neurons exert inhibition of the periaqueductal gray (PAG).

DOI: https://doi.org/10.7554/eLife.47106.013

neurons (*Zhang et al., 1993*), and mice without R type $Ca^{2+}$ channels have deficits in fear response (*Lee et al., 2002*).

In summary, we found that *Ano2*, is expressed in somatostatin-positive GABAergic neurons of central lateral amygdala. TMEM16B-CaCC regulates action potential waveform and GABAergic neurotransmission in these neurons as well as fear and anxiety-like behaviors; the physiological function of TMEM16B is important for expression of anxiety and context-independent fear memory. Better understanding of TMEM16B-CaCC regulation of neuronal signaling could lead to novel strategies for treating anxiety disorders such as panic disorder and post-traumatic stress disorder (PTSD).

## Materials and methods

### Animals

Adult mice aged 12–16 weeks were used for behavioral experiments. *Ano2* knockout mice and control littermates were bred in the on a C57BL6 background as described previously (*Zhang et al., 2017*). To assist electrophysiological recording of somatostatin interneurons, we used the *Sst-IRES-*

*Cre; Ai14* mice expressing Tdtomato in Somatostatin interneurons to fluorescently mark somato-statin expressing cells. Ai14 mice were crossed with *Ano2$^{+/-}$* mice to generate *Ano2$^{+/-}$; Ai14* mice, and further crosses with *Sst-IRES-Cre; Ano2$^{+/-}$; Ai14* mice resulted in fluorescently marking SOM$^+$ cells in *Ano2* KO mice and control mice for slice recording. Mice aged 16–21 days were used for brain slice recording. All mice were socially housed four per cage under a 12 hr light/dark cycle with food and water ad libitum. The use and care of the mice complied with the guidelines of the Institutional Animal Care and Use Committee of UCSF (IACUC protocol AN181236), in accordance with the US National Institute of Health guidelines. Two cohorts were used in the behavioral assays. The first cohort was tested with open field, elevated plus maze, fear conditioning and hot plate. The second cohort was tested with light/dark box and pre-pulse inhibition. There is at least one week between consecutive assays, starting from assays with weak stimulation and followed with tests with progressively stronger stimulation.

## Behavioral assays
### Open Field Test
Mice are placed in the center area of open field chamber (16 in x 16 in) with dim light in the dark phase for 10 min. Mice trajectory was recorded with photobeam assays (San Diego Instrument, San Diego, CA). Center area is a 8 in x eight in square in the center part of the chamber. Anxiety-related behavior was defined by ratio of traveled in the center area to total area.

### Elevated Plus Maze
The elevated plus maze consists of two open and two enclosed arms elevated 63 cm above the ground (Hamilton-Kinder, Poway, CA). Mice are placed at the intersection between the open and closed arms and allowed to explore for 10 min. Anxiety-related behavior was measured by the time and distance traveled in the open arm.

### Light/Dark Box
A dark box with a small opening in the middle is inserted into the open field such that it divides the chamber into two equal portions, a light side and a dark side. Mice are placed inside the dark box in plastic chambers and allowed to explore for 10–30 min (*Cryan and Holmes, 2005*).

### Fear conditioning
#### Training
Mice are placed in the fear conditioning apparatus with light set at 3, fan on, trays sprayed with Windex, and even-barred metal floor grid exposed. Test starts with a 5 min baseline period to access baseline freezing level. Then a combination of a 30 s 80 dB tone that co-terminates with a 2 s, 0.45 mA foot shock is presented three times, separated by 120 s intervals during which freezing is monitored.

#### Context Test
The Context Test takes place 24 hr after Training. Setup is the exactly the same as for training. Mice are placed in the fear conditioning apparatus for 8 min. No shock is presented. Freezing is monitored.

#### Cued Fear Test
The Cued Fear Test takes place 5–24 hr after the Context Test. The context is altered by changing floor grid, turning the fan off and spraying with Simple Green. Mice are placed in the fear conditioning apparatus for 12 min. After a 5 min baseline period four tones are delivered as described in *Training,* separated by 120 s intervals during which freezing is monitored. No shock is presented (*Kopec et al., 2007*).

## Pre-pulse inhibition
### Habituation
Mice are given 5 min to acclimate to the restraining chamber and the 64 dB background noise. After 5 min, mice are introduced to a series of acoustic pulses lasting for 20 min in which weaker acoustic

signal (pre-pulse) is ahead of some pulses at random intervals. Trials with no auditory stimulus are also included to provide a measure of baseline activity. The test is comprised of a total of 80 trials (twenty-four 40 ms/120 dB trials, fourteen each of the 4dbpp/40 ms/120 dB, 15dbpp/40 ms/120 dB, 26dbpp/40 ms/120 dB, and no stim trials). The inter-stimulus interval (ISI) is variable with a mean of 15 s, and a range of 8–22 s.

### Data extraction

We calculated the mean startle amplitude for each trial type: 40 ms/120 dB, 4dbpp/40 ms/120 dB, 15dbpp/40 ms/120 dB, 26dbpp/40 ms/120 dB, and no stimulus across the entire session. To calculate the % pre-pulse inhibition, we used this formula: 100 * ((mean of 40 ms/120 dB trials-mean of pre-pulse trial type)/(mean of 40 ms/120 dB)) (*Vitucci et al., 2016*).

## In situ hybridization

Mice were euthanized with isoflurane and mice brains were removed without perfusion. The specimen was frozen on dry ice or in liquid Nitrogen within 5 min of tissue harvest. Frozen Brain was embedded in cryo-embedding medium (OCT), and then sectioned with a Cryostat. Sections were fixed with chilled 4% PFA for 15 min, and then washed with 0.1 M PBS twice. Sections were then dehydrated with 50% ethanol, 70% ethanol, and 100% ethanol sequentially. Slides were treated with protease and then incubated with customized probe for 2 hr. Signal was amplified by using the ACD RNAscope Fluorescent Multiplex Kit. RNAscope Probe-Mm-Ano2 (Cat No. 441951), Probe-Mm-Gad2-C3 (Cat No. 439371-C3), Probe-Mm-Sst (Cat No. 404631-C2). Image is analyzed by Fuji.

## Slice preparation

Slices (300 μm) were prepared as previously described (*Li et al., 2012*). In brief, brain slices from P16-P20 were sliced with a Vibroslice (Leica VT 1000S) in ice-cold artificial cerebrospinal fluid (ACSF), which consisted of (in mM) 125 NaCl, 3 KCl, 1.25 $NaH_2PO_4$, 1.3 $MgCl_2$, 2.6 $CaCl_2$, 25 $NaHCO_3$ and 10 glucose. Slices were recovered for ~30 min at 33°C, incubated in ACSF for ~60 min at 22°C, then transferred to the recording chamber and perfused (3 ml min$^{-1}$) with ACSF at 22°C. All external solutions were saturated with 95% $O_2$, 5% $CO_2$.

## Electrophysiology

Neurons were visualized with a 60x water-immersion lens (Zeiss) and recorded using whole-cell techniques (MultiClamp 700B Amplifier, Digidata 1322A analog-to-digital converter) and pClamp 9.2 software (Axon Instruments/Molecular Devices). The solution for action potential recording containing (in mM): 125 potassium gluconate, 15 KCl, 10 HEPES, 4 Mg-ATP, 0.3 Na-GTP, 10 disodium phosphocreatine and 0.2 EGTA, pH 7.2 with KOH, 288 mOsm. Voltage threshold was defined as $dV/dt = 10$ mV ms$^{-1}$. Spike amplitude was measured as the voltage difference between the peak amplitude and the threshold of the action potential. After afterhyperpolarization potential (AHP) was measured as the difference between the spike threshold and the minimum voltage after the action potential peak. Spike half width was measured at half the spike amplitude. APD90% was defined as width of the action potential repolarization to 90%. All analysis was performed using pClamp/Clampfit.

To measure the calcium activated chloride currents in voltage clamp, the patch pipette (3–4 MΩ) was filled with a solution (*Zhang et al., 2017*) containing (in mM): 110 Cs-methanesulphonate, 20 tetraethylammonium chloride (TEACl), 8 KCl, 10 HEPES, 4 MgATP, 0.3 NaGTP, 0.2 EGTA, 5 QX-314-Br, pH 7.25 with CsOH, 290 mOsm. For pharmacological experiments, 200 μM $Cd^{2+}$ or 100 μM niflumic acid (NFA) was applied to ACSF. The holding potential for CaCC recordings was −80 mV.

To record evoked postsynaptic currents, the patch pipette (3–4 MΩ) was filled with a solution same as the one for CaCC recording. The puff micropipette (4–5 MΩ) was placed 20–40 μm away from the neuron recorded. A low gas pressure (4–6 psi) was applied to the puff micropipette containing GABA by Picrospritzer. 100 μM GABA is dissolved in ACSF freshly. The holding potential for eIPSC recording is 0 mV.

To record GABAergic postsynaptic currents in voltage clamp, a high chloride internal solution (in mM) was used: 110 CsCl, 30 Potassium gluconate, 10 HEPES, 2 $MgCl_2$, 4 Mg-ATP, 0.3 Na-GTP 10 disodium phosphocreatine and 0.2 EGTA, pH 7.2 with KOH, 288 mOsm. IPSCs were recorded with

50 µM DL-AP5 and 20 µM DNQX in ACSF. mIPSCs were recorded in the presence of 50 µM DL-AP5, 20 µM DNQX and 1 µM TTX. The holding potential for both action potential and IPSCs recordings was −70 mV. The threshold amplitude for mIPSCs is 15 pA. As most mIPSCs are less than 40 pA in amplitude, we analyzed all the events in sIPSC recordings that are greater than 40 pA, hence unlikely to arise from spontaneous GABA release. During recording, access resistance was compensated by up to 80%. Neurons with series resistance above 20 MΩ and >20% of changes throughout the recording were discarded, otherwise were used for analysis. Bridge balance in current clamp and whole-cell compensation in voltage clamp were adjusted throughout the recording. Recordings were sampled at 100 kHz for action potentials, sampled at 10 kHz for IPSCs and CaCC recordings and filtered at 3 kHz. IPSCs were analyzed using Mini Analysis Program (Synaptosoft).

## Quantification and statistical analysis

All data are represented as mean ± SEM. All statistical analyses were performed with Prism (Graph-Pad) with appropriate method which indicated in the figure legend. Paired or unpaired t tests and Mann Whitney tests were used for two group comparisons. Two-way repeated-measures ANOVA was used to perform group comparisons. Data are considered to be statistically significant if $p < 0.05$. Model illustration is created by https://biorender.com/.

## Reagents

Pharmacological drugs used included 100 µM Picrotoxin (Tocris, Cat. No. 1128/1G), 50 µM DL-AP5 (Tocris, Cat. No. 0105/10), 20 µM DNQX (Tocris, Cat. No. 2312/10), Tetrodotoxin (abcam, ab120054), QX-314 (Tocris, Cat. No. 1014/100), Cadmium chloride (Sigma Aldrich, 10108-64-2), Niflumic acid (Sigma Aldrich, 4394-00-7) and 100 µM GABA (Sigma Aldrich, A2129).

## Acknowledgements

We thank for Dr. Andrea Hasenstaub for providing Som-ires-Cre and Ai14 mice, Dr. Chin Fen Teo, Dr. Yanmeng Guo, Dr. Tun Li for stimulating discussions and critical comments. This study is supported by the NIH RO1 grant NS069229 to LYJ and the F32HD089639 to MH. YNJ and LYJ are Howard Hughes Medical Institute investigators.

## Additional information

### Funding

| Funder | Grant reference number | Author |
| --- | --- | --- |
| National Institute for Health Research | RO1 NS069229 | Lily Yeh Jan |
| Eunice Kennedy Shriver National Institute of Child Health and Human Development | F32HD089639 | Mu He |
| Howard Hughes Medical Institute | | Yuh Nung Jan |

The funders had no role in study design, data collection and interpretation, or the decision to submit the work for publication.

### Author contributions

Ke-Xin Li, Conceptualization, Formal analysis, Investigation, Methodology, Writing—original draft, Writing—review and editing; Mu He, Investigation, Writing—review and editing; Wenlei Ye, Methodology; Jeffrey Simms, Investigation, Behavioral studies; Michael Gill, Resources, Methodology, Behavioral studies; Xuaner Xiang, Investigation, Assisted with genotype, brain slice cutting and mouse data analysis during manuscript revision; Yuh Nung Jan, Resources, Writing—review and editing; Lily Yeh Jan, Resources, Supervision, Funding acquisition, Writing—review and editing

## Author ORCIDs

Ke-Xin Li (ID) https://orcid.org/0000-0003-3879-294X
Wenlei Ye (ID) http://orcid.org/0000-0002-4694-1493
Yuh Nung Jan (ID) http://orcid.org/0000-0003-1367-6299
Lily Yeh Jan (ID) https://orcid.org/0000-0003-3938-8498

## Ethics

Animal experimentation: The use and care of the mice complied with the guidelines of the Institutional Animal Care and Use Committee of UCSF (IACUC protocol AN181236), in accordance with the US National Institute of Health guidelines.

## Decision letter and Author response

Decision letter https://doi.org/10.7554/eLife.47106.016
Author response https://doi.org/10.7554/eLife.47106.017

## Additional files

### Supplementary files

• Transparent reporting form
DOI: https://doi.org/10.7554/eLife.47106.014

### Data availability

All data generated or analysed during this study are included in the manuscript and supporting files.

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
