## [Decision Letter]

Thank you for submitting your article "TMEM16B regulates anxiety-related behavior and GABAergic neuronal signaling in the central lateral amygdala" for consideration by *eLife*. Your article has been reviewed by three peer reviewers, and the evaluation has been overseen by a Reviewing Editor and Catherine Dulac as the Senior Editor. The reviewers have opted to remain anonymous.

The reviewers have discussed the reviews with one another and the Reviewing Editor has drafted this decision to help you prepare a revised submission.

Summary:

While the reviewers were overall positive about the fundamental contribution this manuscript makes to the neurobiology of TMEM16B (and Calcium-activated Chloride currents in general), they raised some substantive concerns especially in regards to the overarching nature of the claims on the relevance of these findings to anxiety related behavior and panic disorder. This emphasis should be significantly toned down in the text. In addition, the reviewers agreed that the synaptic phenotype associated with TMEM16B function should be clarified.

Essential revisions:

1) The authors need to substantially tone down the discussion on panic disorders. Inclusion of the caveats discussed in the reviews within the text of the manuscript will help readers evaluate the findings (reviewer #1, point 2 and reviewer #3, point 4).

2) The manuscript should incorporate more evidence on the nature of the synaptic change observed to establish that it is indeed a presynaptic effect.

3) Please include data demonstrating that Calcium-activated Chloride currents are missing in non-SOM+ neurons in the CeA to confirm cellular specificity.

Reviewer #1:

The authors show that knockout of TMEM16B, a calcium activated chloride channel localized in somatostatin positive subgroup of neurons of the central lateral nucleus of amygdala, results in reduced anxiety-related behaviors in a battery of behavioral tests as well as changes in the electrophysiological properties of the aforementioned cells and the overall circuitry. The calcium activated chloride current was beautifully characterized electrophysiologically. Previous work from the same lab characterized TMEM16B current in the hippocampus (Huang et al., 2012). Similar to previous findings elicited with calcium activated chloride current inhibitors NFA and NPPB, TMEM16B KO caused widening of the action potential resulting in increased frequency and amplitude of action potential dependent spontaneous IPSCs. Even though the observed electrophysiological changes are not tied to the observed behavioral changes, this is a well-written manuscript presenting novel and interesting data obtained from well-designed experiments.

1) The authors characterize the anxiolytic-like effect of TMEM16B KO in OFT, EPM and light/dark box paradigms. However, the knockout is global in the mice model used and involvement of other brain areas, especially hippocampus (Kjelstrup et al., 2002), may be a confounding factor. A more targeted knockout approach in SOM+ neurons of CeL or a rescue experiment of TMEM16B in the Cl in the KO line would strengthen the claims of the paper.

2) In the second paragraph of the Introduction, the authors cited a GWAS study (Otowa et al., 2009) to support the rationale that altered function of TMEM16B may be involved in the pathogenesis of panic disorder (PD). The same group published a follow-up paper a year later, stating that they were unable to find any significant association between previously reported single nucleotide polymorphisms (SNPs), including the one reported in TMEM16B, and PD (Otowa et al., 2010). Similarly, a meta-analysis of GWAS studies by the same group published in 2012 was failed to show any association between SNPs in TMEM16B and PD (Otowa et al., 2012). Given the more recent papers with a greater number of subjects found no association between SNPs of TMEM16B and PD, the authors should rewrite the paper accordingly.

3) The findings related to the miniature release are, indeed, interesting; however the importance and/or pathophysiological relevance of these findings don't come to a conclusion other than supporting that TMEM16B can modulate the release at presynaptic terminal. The authors should limit their discussion of findings in miniature release in the sixth paragraph of the Discussion, since they don't present enough data to implicate mIPSCs in TMEM16 KO phenotype. The current finding is itself intriguing since the previous work from the same lab did not report any presynaptic action of TMEM16B in hippocampus (Huang et al., 2012). In addition, the source of postsynaptic calcium to activate TMEM16B should be discussed in greater detail.

Kjelstrup, K. G., F. A. Tuvnes, H. A. Steffenach, R. Murison, E. I. Moser, and M. B. Moser. 2002. 'Reduced fear expression after lesions of the ventral hippocampus', Proc Natl Acad Sci U S A, 99: 10825-30.

Reviewer #2:

This is a fairly simple and solid paper that shows a very interesting behavioral phenotype in TMEM16B deletion mice. The set of behaviors is really extremely interesting. The electrophysiology too is performed to a high standard. Overall, this is a solid study with clear and compelling data.

The authors have made a connection to panic disorder (PD) perhaps as a way to increase the appeal of the study. This is a personal choice, but there appears to be nothing wrong with just reporting the results as a TMEM16B knock-out paper and discussing the possible relevance to PD in the Discussion. This may be better than mentioning PD in the Abstract and Introduction, which as the authors point out later is a tenuous connection as the SNP in TMEM16B may lead to increased activity as the link to the phenotype shown. However, these are clearly very sophisticated/accomplished authors and so it is really up to them how they write it. If they do lead with PD, then I think they need to expand the second paragraph of the Discussion, and explicitly state the limitations of the work.

Reviewer #3:

The manuscript by Li and co-workers describes the effects of TMEM16B deletion on anxiety and fear-related behaviors and the authors show an anxiolytic-like profile in KO mice. They then localize expression of this protein to CeAL SOM+ neurons and demonstrate its functional existence in these cells and the effects of global deletion to increase AP 1/2 width. They then go on to demonstrate increased spontaneous GABAergic, but reduced miniature GABAergic signaling, onto CeAL SOM+ neurons. The strengths of the work include a clear and robust behavioral phenotype in the KO mice, elegant demonstration of functional expression within CeA SOM+ neurons, and elucidation of the significance of the global TMEM16B deletion for CeA GABAergic signaling. However, there are a number of significant limitations that reduce the strength of the conclusions that can be drawn, which reduces the overall impact of the work. My major conceptual issues are as follows:

1) Both frequency and amplitude of sIPSCs are altered in KO mice, yet the authors posit an increase in the number of release sites/docked vesicles to explain these data (Figure 8), which they posit to be related to increases in AP width. Exclusion of postsynaptic changes in GABA receptor function via local application would be needed to support the conclusion that the changes in sIPSC amplitude are not mediated by postsynaptic alterations.

2) The model posits SOM-SOM inhibitory connections are enhanced in KO mice; however, the electrophysiological approaches do not allow for identification of the afferent source of the GABAergic input onto these SOM neurons being recorded from. They are likely a mixture of PKCd and CRF or other inputs in addition to local SOM-SOM connections.

3) The effects of global deletion are not localized to CeA or even SOM neurons. Without this the cellular/synaptic mechanisms by which TMEM16B deletion causes observed behavioral phenotypes is not advanced.

4) Panic-like behaviors are not examined. While anxiety is a component of panic disorder, animal models that much more closely relate to core panic symptoms in humans are available (i.e. hypercapnia-induced behaviors).

5) Negative control studies are missing. For example, can the authors show lack of functional CaCC in non-SOM+ neurons in the CeA to confirm cellular specificity.

---

## [Author Response]

Essential revisions:1) The authors need to substantially tone down the discussion on panic disorders. Inclusion of the caveats discussed in the reviews within the text of the manuscript will help readers evaluate the findings (reviewer #1, point 2 and reviewer #3, point 4).

We removed the statement regarding association of TMEM16B with panic disorder from Introduction and toned down the discussion about the potential relevance of TMEM16B mutation to anxiety and fear related diseases in the Discussion. We have also incorporated discussions about the caveats raised by the reviewers.

2) The manuscript should incorporate more evidence on the nature of the synaptic change observed to establish that it is indeed a presynaptic effect.

We found that loss of TMEM16B function led to an increase of sIPSC amplitude and frequency (Figure 6). As both the amplitude and frequency of sIPSC are increased in KO neurons, we considered two possibilities. First, the effect may reflect increased presynaptic transmitter release. Second, it may arise from both increased presynaptic transmitter release and increased postsynaptic response. To discriminate between these two possibilities, we evaluated GABA receptor function of SOM+ neurons. We performed additional experiments to show that the eIPSC amplitude is comparable between control and TMEM16B KO neurons in Figure 6—figure supplement 1, excluding the involvement of GABA receptor modulation by TMEM16B. Taken together, our findings indicate that the potentiation of sIPSC could have arisen from the broadening of action potentials in presynaptic neurons lacking TMEM16B.

3) Please include data demonstrating that Calcium-activated Chloride currents are missing in non-SOM+ neurons in the CeA to confirm cellular specificity.

We followed the advice and incorporated additional experiments to show an unremarkable Calcium activated Chloride currents in non-SOM+ neurons, consistent with our finding that TMEM16B-CaCCs is expressed on SOM+ neurons specifically.

Reviewer #1:[…] 1) The authors characterize the anxiolytic-like effect of TMEM16B KO in OFT, EPM and light/dark box paradigms. However, the knockout is global in the mice model used and involvement of other brain areas, especially hippocampus (Kjelstrup et al., 2002), may be a confounding factor. A more targeted knockout approach in SOM+ neurons of CeL or a rescue experiment of TMEM16B in the Cl in the KO line would strengthen the claims of the paper.

Pursuing these experiments would require much longer than the timeframe allowed for revising our *eLife* manuscript. We have revised our manuscript to spell out clearly that additional experiments as outlined by the reviewer will be necessary to assess the contribution of TMEM16B in amygdala to the behavioral phenotypes uncovered in this study.

2) In the second paragraph of the Introduction, the authors cited a GWAS study (Otowa et al., 2009) to support the rationale that altered function of TMEM16B may be involved in the pathogenesis of panic disorder (PD). The same group published a follow-up paper a year later, stating that they were unable to find any significant association between previously reported single nucleotide polymorphisms (SNPs), including the one reported in TMEM16B, and PD (Otowa et al., 2010). Similarly, a meta-analysis of GWAS studies by the same group published in 2012 was failed to show any association between SNPs in TMEM16B and PD (Otowa et al., 2012). Given the more recent papers with a greater number of subjects found no association between SNPs of TMEM16B and PD, the authors should rewrite the paper accordingly.

We have rewritten the paper, removed the statement about association of TMEM16B with panic disorder from Abstract and Introduction. In the revised manuscript, we start with reporting TMEM16B knockout phenotype and discuss the potential relevance of TMEM16B mutation with anxiety and fear related diseases in the Discussion.

3) The findings related to the miniature release are, indeed, interesting; however the importance and/or pathophysiological relevance of these findings don't come to a conclusion other than supporting that TMEM16B can modulate the release at presynaptic terminal. The authors should limit their discussion of findings in miniature release in the sixth paragraph of the Discussion, since they don't present enough data to implicate mIPSCs in TMEM16 KO phenotype. The current finding is itself intriguing since the previous work from the same lab did not report any presynaptic action of TMEM16B in hippocampus (Huang et al., 2012). In addition, the source of postsynaptic calcium to activate TMEM16B should be discussed in greater detail.

We followed the advice of the reviewer and deleted the discussion about mIPSCs arising from spontaneous release. We added discussion of the possible source of calcium in postsynaptic component to activate TMEM16B-CaCC in the revised manuscript.

Reviewer #3:The manuscript by Li and co-workers describes the effects of TMEM16B deletion on anxiety and fear-related behaviors and the authors show an anxiolytic-like profile in KO mice. They then localize expression of this protein to CeAL SOM+ neurons and demonstrate its functional existence in these cells and the effects of global deletion to increase AP 1/2 width. They then go on to demonstrate increased spontaneous GABAergic, but reduced miniature GABAergic signaling, onto CeAL SOM+ neurons. The strengths of the work include a clear and robust behavioral phenotype in the KO mice, elegant demonstration of functional expression within CeA SOM+ neurons, and elucidation of the significance of the global TMEM16B deletion for CeA GABAergic signaling. However, there are a number of significant limitations that reduce the strength of the conclusions that can be drawn, which reduces the overall impact of the work. My major conceptual issues are as follows:1) Both frequency and amplitude of sIPSCs are altered in KO mice, yet the authors posit an increase in the number of release sites/docked vesicles to explain these data (Figure 8), which they posit to be related to increases in AP width. Exclusion of postsynaptic changes in GABA receptor function via local application would be needed to support the conclusion that the changes in sIPSC amplitude are not mediated by postsynaptic alterations.

We thank the reviewer for the helpful suggestion. We measured current response to application of GABA to SOM+ neurons, now shown in Figure 6—figure supplement 1.

2) The model posits SOM-SOM inhibitory connections are enhanced in KO mice; however, the electrophysiological approaches do not allow for identification of the afferent source of the GABAergic input onto these SOM neurons being recorded from. They are likely a mixture of PKCd and CRF or other inputs in addition to local SOM-SOM connections.

We thank the reviewer for raising this point. We agree that the afferent GABAergic inputs onto SOM+ neurons are a mixture of SOM+ and non-SOM+ neurons. We have revised the model shown in Figure 8of the revised manuscript.

3) The effects of global deletion are not localized to CeA or even SOM neurons. Without this the cellular/synaptic mechanisms by which TMEM16B deletion causes observed behavioral phenotypes is not advanced.

We acknowledge that with global deletion we cannot exclude the involvement of TMEM16B in neurons other than SOM+ neurons in CeA in the behavioral phenotypes, and we discuss this point in the revised manuscript.

4) Panic-like behaviors are not examined. While anxiety is a component of panic disorder, animal models that much more closely relate to core panic symptoms in humans are available (i.e. hypercapnia-induced behaviors).

We thank the reviewer for raising this point. In the revised article, we followed the suggestion of the editors and reviewers and deemphasize the association of TMEM16B with panic disorder. Instead, we start with reporting the TMEM16B knockout phenotype and discuss its potential relevant to anxiety and panic disorders in the Discussion.

5) Negative control studies are missing. For example, can the authors show lack of functional CaCC in non-SOM+ neurons in the CeA to confirm cellular specificity.

We have followed the advice of the reviewer and added a new Figure 4—figure supplement 1 to show that functional CaCC current is missing in non-SOM+ neurons.